# Interactions between S100A9 and Alpha-Synuclein: Insight from NMR Spectroscopy

**DOI:** 10.3390/ijms23126781

**Published:** 2022-06-17

**Authors:** Zigmantas Toleikis, Raitis Bobrovs, Agne Janoniene, Alons Lends, Mantas Ziaunys, Ieva Baronaite, Vytautas Petrauskas, Kristine Kitoka, Vytautas Smirnovas, Kristaps Jaudzems

**Affiliations:** 1Latvian Institute of Organic Synthesis, Aizkraukles 21, LV-1006 Riga, Latvia; raitis.bobrovs@osi.lv (R.B.); alons.lends@gmail.com (A.L.); kitoka@osi.lv (K.K.); kristaps.jaudzems@osi.lv (K.J.); 2Life Sciences Center, Institute of Biotechnology, Vilnius University, Saulėtekio 7, 10257 Vilnius, Lithuania; agne.vegyte@bti.vu.lt (A.J.); mantas.ziaunys@gmc.vu.lt (M.Z.); ieva.baronaite@chgf.stud.vu.lt (I.B.); vytautas.petrauskas@bti.vu.lt (V.P.); vytautas.smirnovas@bti.vu.lt (V.S.)

**Keywords:** S100A9, synuclein, amyloid proteins, fibrils, NMR, ThT fluorescence assay, AFM, FTIR, MTT, LDH cell toxicity tests

## Abstract

S100A9 is a pro-inflammatory protein that co-aggregates with other proteins in amyloid fibril plaques. S100A9 can influence the aggregation kinetics and amyloid fibril structure of alpha-synuclein (α-syn), which is involved in Parkinson’s disease. Currently, there are limited data regarding their cross-interaction and how it influences the aggregation process. In this work, we analyzed this interaction using solution ${}^{19}$F and 2D ^15^N–^1^H HSQC NMR spectroscopy and studied the aggregation properties of these two proteins. Here, we show that α-syn interacts with S100A9 at specific regions, which are also essential in the first step of aggregation. We also demonstrate that the 4-fluorophenylalanine label in alpha-synuclein is a sensitive probe to study interaction and aggregation using ${}^{19}$F NMR spectroscopy.

## 1. Introduction

One of the causes of neurodegenerative disorders and other amyloidoses is the accumulation of amyloid fibrils in the brain or other tissues [1,2]. Currently, more than 30 known proteins are associated with these disorders [3]. An increasing number of people suffering from amyloid-caused dysfunctions [4], and a lack of medicines [5,6] make the study of amyloid protein aggregation very important. Incomplete knowledge about the structures of amyloid aggregation intermediates and pathways to target is the key factors limiting drug discovery against amyloid-related diseases [7].

Alpha-synuclein (α-syn) is an intrinsically disordered protein forming clusters of aggregates, and their accumulation is associated with the onset of Parkinson’s disease. The structure of α-syn contains three segments: N-terminal, non-beta-amyloid component (NAC), and C-terminal. The N-terminal sequence of α-syn forms an alpha-helix when it interacts with lipid surfaces. The NAC segment forms beta strands during fibril formation. The C-terminal part is negatively charged and can interact with the N-terminal part, NAC segment, [8] and bind divalent metal ions. The C-terminal part of α-syn with the bound calcium ions can influence aggregation kinetics as well as interaction with lipid vesicles [9] and membranes [10,11]. In its native state, α-syn contributes to the stabilization of vesicles, which transport neurotransmitters in the pre-synaptic part of neurons [12]. However, the physiological function is lost when α-syn forms aggregates, large neuronal or glial cytoplasmic inclusions [13,14,15].

Different structure α-syn fibrils have been observed, when the protein was aggregated under various conditions in vitro [16,17].

The fibril structure is related to distinct neurodegenerative diseases [18,19]. The aggregation of α-syn is a stochastic process, which depends on the experimental conditions such as solution pH [20], ionic strength [21,22], protein concentration [23], or the presence of other amyloid proteins [24,25,26,27,28], including S100A9 [29] or non-amyloid proteins [30,31,32]. Furthermore, 65 proteins in Lewy’s bodies were shown to interact with α-syn [33]. In some cases, this interaction can be transient and protective from aggregation as was shown with the small heat shock proteins interacting with α-syn [32].

S100A9 is a calcium-binding inflammatory protein [33,34] that is mainly produced in neutrophiles [35] and to some extent in neurons and microglia. The structure of S100A9 is composed of four alpha-helices forming two calcium-binding helix–loop–helix motifs, known as EF-hands, where one calcium ion is coordinated in each loop. The first EF-hand includes Helix 1 (residues Gln 7 to Ser 23), calcium-binding loop (Val 24–Asn 33) and Helix 2 (Gln 34–Asp 44). The hinge region from Leu 45 to Asn 55 binds the second EF-hand composed of Helix 3 (Glu 56–Leu 66), calcium-binding loop (Asp 67–Ser 75) and Helix 4 (Phe 76–Arg 85) [36]. Helix 4 ends at the residues 85–87 in crystal structures (1IRJ, 4GGF, 5W1F) or at Met 94 in solution structure (5I8N) [36,37,38,39]. S100A9 can form amyloid fibrils in vitro [40] and in vivo [41,42], which might be related to amyloidoses [43,44]. S100A9 can speed up the aggregation of α-syn [29] and stabilize the particular population of α-syn fibrils, which have different secondary structure motifs and amyloid-specific dye-binding properties [45]. S100A9 can change the aggregation of amyloid-beta in vitro [46] and is also found co-aggregated in the brain tissues in the case of Alzheimer’s disease [41]. In order to understand the nature of how S100A9 influences amyloid aggregation of other proteins, it is necessary to understand the possible interaction of these proteins.

Nuclear magnetic resonance (NMR) spectroscopy is a powerful technique for studying protein structures and interactions at an atomic resolution. ${}^{19}$F is a very sensitive probe for NMR spectroscopy, which enables studying the conformational changes of protein and interactions at specific labeled sites with high sensitivity similar to ${}^{1}$H [47]. Usually, the aromatic amino acid residues are labeled with ${}^{19}$F, which was shown not to affect the structure and stability of the protein [48]. However, in some cases, the high level of fluorine labeling can affect protein properties [49]. Two-dimensional (2D) ^15^N–^1^H HSQC NMR spectroscopy allows determining the fingerprint of ${}^{15}$N-labeled protein and follows residue-specific changes during protein modification, unfolding, aggregation or interaction. The solid-state NMR (ssNMR) spectroscopy has been widely applied to analyze α-syn amyloid fibrils [17,50,51,52].

In this work, we analyzed the interaction of S100A9 with α-syn using 1D ${}^{19}$F and 2D ^15^N–^1^H HSQC NMR spectroscopy and molecular dynamics simulations. In addition, the aggregation kinetics, fibril structure, and toxicity of α-syn were analyzed by ThT fluorescence assay, atomic force microscopy (AFM), Fourier transformed infrared (FTIR) spectroscopy, solution and solid-state NMR spectroscopy, cell viability and membrane permeability tests. The study revealed that S100A9 interacts with α-syn, influencing the aggregation process and the resulting fibril structure and toxicity.

## 2. Results

### 2.1. α-syn Binding with S100A9

To study the interaction between α-syn and S100A9 by ${}^{19}$F NMR spectroscopy, two 4-fluorophenylalanines (4F-Phe) were introduced into α-syn Phe 4 and Phe 94 positions, which allowed following the changes in the N-terminal region and the NAC domain. The 1D ${}^{19}$F NMR spectrum of α-syn showed two separate peaks corresponding to 4F-Phe 4 and 4F-Phe 94. The position and separation of these peaks depended on temperature, and good separation was observed when the temperature was lower than 20 ${}^{\circ }$C (Figure 1A). Addition of the same molar concentration (50 μM) of S100A9 to α-syn solution changed the intensity of one peak. Despite the fact that peaks of Phe 4 and Phe 94 were more separated at 10 ${}^{\circ }$C or 5 ${}^{\circ }$C, the largest intensity decrease after the addition of S100A9 was observed at 15 ${}^{\circ }$C (Figure 1A). The spectrum of F4Y α-syn mutant showed that the first peak at −116.20 ppm corresponded to labeled Phe 4 and the second peak at −116.22 ppm corresponded to Phe 94 (Figure 1B).

The addition of 500 μM S100A9 to 50 μMα-syn solution without calcium shifted Phe 4 peak to the left and intensity decreased slightly, indicating that the N-terminal part of α-syn interacts with S100A9 (Figure 1C). The addition of 5 mMCaCl_2_ to the α-syn solution did not change the positions and intensities of peaks, indicating that calcium does not bind to α-syn close to Phe 4 or Phe 94 and does not affect the environment around Phe 4 and Phe 94. However, when 500 μM S100A9 was present in 50 μMα-syn solution containing 5 mM CaCl_2_, the intensity of Phe 4 peak decreased more than twice (Figure 1D). This observation indicates that calcium enhances the interaction of α-syn with S100A9. The titration analysis showed that the intensity decreased and the line width increased for Phe 4 peak when the concentration of S100A9 was increased up to 35 μM, 70 μM, 125 μM, 250 μM or 500 μM (Figure 1D,E).

To confirm the ${}^{19}$F NMR results and identify the other interaction sites, also on S100A9, 2D ^15^N–^1^H HSQC NMR experiments were performed using uniformly ${}^{15}$N-labeled protein samples. The ^15^N–^1^H HSQC spectra of α-syn were recorded at 10 ${}^{\circ }$C and 15 ${}^{\circ }$C, as some of the overlapping peaks were better resolved at one distinct temperature (for instance, Phe 4, Lys 12, Glu 28, Ala 29, Gln 79, and Leu 113 were better resolved at 15 ${}^{\circ }$C). However, the analysis of intensity and chemical shift changes gave similar results at both temperatures (Figure 2 and Figure A2A,B). The ^15^N–^1^H HSQC spectra of α-syn were almost the same in the absence and presence of S100A9, as there was no clear chemical shift changes of the peaks (Appendix A
Figure A1A). Nevertheless, peak intensity changes of α-syn were observed in three regions: (1) the N-terminal part starting from Val 3 (the first residue which is assigned) and ending at Ser 9; (2) the region from Glu 35 to Gly 41 and (3) to a less extent in the C-terminal part starting from Glu 123 and ending at Tyr 133 (Figure 2). Small chemical shift changes were also noticed in these regions. The most affected amino acid residues were Val 3, Phe 4, Met 5, Lys 6, Leu 8, Gly 36, Val 37, Leu 38, Thr 39, Val 40, and Gly 41. A very clear but small shift and decrease of intensity was observed for the Met 5 peak, which was very well separated (not overlapping). Additionally, the residue Asn 65 outside the described regions showed clear intensity and chemical shift changes. A similar result was observed when S100A9 concentration was five times lower (1:1 molar ration of α-syn and S100A9), but the region around Tyr 39 was less pronounced (Figure A2C,D).

The ^15^N–^1^H HSQC spectra of S100A9 in the presence and absence of α-syn were very similar to each other, except for a clear intensity decrease of the peaks (Apendix Figure A1B). The detailed analysis showed that the intensity decreased for almost all S100A9 peaks in the presence of α-syn (Figure 3A). When 100 μM S100A9 was in the solution containing 125 μMα-syn, the largest intensity decrease was observed in a few regions: (1) Leu 8, Arg 10, Asn 11, Asn 17 and Phe 19 in Helix 1, (2) Leu 40 in Helix 2, (3) Phe 48 and Glu 52 in a hinge region, (4) Glu 56 (a clear intensity decrease was evident at 30 ${}^{\circ }$C, as the peak intensity at 25 ${}^{\circ }$C was too low, even without addition of α-syn), Glu 60, Met 63, and Glu 64 in Helix 3, (5) Lys 72 and Ser 75 in the loop close to Helix 4 (Figure 3A). The most pronounced region, which showed a large intensity decrease, was in the C-terminal part composed of Leu 86, Thr 87, Trp 88, Ala 89, and Ser 90. The higher chemical shift perturbations were also observed in this region (Figure 3B). The intensity of S100A9 decreased more, when the concentration of added α-syn increased, to reach a 1:5 molar ratio of S100A9:α-syn (Figure 3C). The CSPs were also higher of the same peaks (Figure 3D).

### 2.2. Molecular Dynamics Simulation of the α-syn-S100A9 Complex

The probable α-syn and S100A9 interactions were further studied using molecular dynamics (MD) simulations. Since there are no α-syn-S100A9 complex structures available, the structure used in MD simulations was derived from the complex of α-syn with the structurally similar protein–calmodulin (PDB ID: 2M55). Despite 11% sequence identity and 32% similarity compared S100A9 with calmodulin, these proteins have a very similar secondary structure (Figure 4A). Both proteins have two EF-hand motifs and similar 3D structures (Figure 4B). We selected a calmodulin complex with sequence truncated α-syn as a template due to the mentioned high similarity of calmodulin and S100A9 secondary structures (Figure 4A,B), and because in the α-syn-calmodulin complex, the key inter-protein interactions are between the N-terminal part of α-syn and central part of calmodulin; similar results were suggested by the NMR spectroscopy data for our system. Since only the N-terminal part of α-syn showed measurable intensity and chemical shift perturbations, only the first 50 amino acids of the α-syn were considered in our simulations to reduce the computational cost (Figure 4C).

The potential α-syn-S100A9 complex structures were sampled by running 20 parallel 100 ns MD simulations. The first half of each simulation was discarded to account for system equilibration, resulting in 1 μs long equilibrated trajectory. MD data revealed that the α-syn-S100A9 complex did not disassemble during the simulations, but the complex was highly flexible. S100A9 maintained its secondary structure throughout simulations, whereas α-syn was very flexible, and the α-helical motif was maintained only at the N-terminal part (residues 1–10), which interacted with S100A9. The α-syn amino acid residues further away from the N-terminal part showed high conformational flexibility. To characterize the dominant α-syn-S100A9 complex structures, clustering was performed on concatenated trajectories, considering both proteins (Figure 4D). The cut-off value of 0.3 nm was used for GROMOS clustering [53]. For each cluster, an inter-residue contact map was generated (up to 1.0 nm inter-residue distance). Information on the 10 most populated clusters is summarized in Supplement Material Figure A3. Five clusters (1, 7–10) of the ten most populated ones were in close agreement with NMR data (Figure 4D). Thus, the MD simulations confirmed interacting regions on α-syn (based on NMR data): (1) N-terminal up to Ser 9, (2) residues 35–41 and on S100A9: (1) residues 8–19, 48–52, 59–72 (with some exceptions) and (2) residues 86–90.

### 2.3. Aggregation of α-syn by ThT Assay

To investigate the influence of S100A9 on α-syn aggregation behavior, we performed α-syn aggregation kinetics by ThT assay. The α-syn aggregation was performed in four different conditions: (**A**)

α-syn without CaCl_2_ and S100A9, (**B**) α-syn without CaCl_2_ but with S100A9, (**C**) α-syn with 1 mMCaCl_2_ but without S100A9, (**D**) α-syn with 1 mM CaCl_2_ and 100 μM S100A9. The spontaneous aggregation of α-syn was very stochastic between repeats of each sample at 37 ${}^{\circ }C$, 300 RPM in respect to ThT maximum fluorescence intensity ($I_{max}$) and the midpoint (half time) of aggregation transition ($t_{50}$) (Figure 5). The highest $I_{max}$ values were observed for samples **A** and **C**. The lower $I_{max}$ values were observed for the samples **B** and **D**, which contained α-syn and S100A9. The double aggregation transition (three states) were observed in one-third of the curves for α-syn samples, which contained S100A9 (samples **B** and **D**). The determined values of $t_{50}$ were broadly distributed for all samples (Figure 5F). The $t_{50}$ values were slightly higher for samples **B** and **D** compared to samples **A** and **D**. Some correlation between $I_{max}$ and $t_{50}$ was observed just in sample **B** and **D** (Figure A4). To conclude these observations, it seems that S100A9 might change the pathway of α-syn aggregation.

### 2.4. Aggregation of α-syn by NMR Spectroscopy

The aggregation of α-syn was also analyzed by 1D ${}^{19}$F and 2D ^15^N–^1^H HSQC NMR spectroscopy. We studied sample conditions **A**–**D**, as described in ThT assay. ${}^{19}$F NMR spectra of 4F-Phe labeled α-syn showed different behavior for the Phe 4 and Phe 94 peaks during aggregation. Samples **A** and **C** were stable up to 13 h at aggregation conditions, showing only a small intensity decrease and a shift of Phe 4 peak (Figure 6A,C). The peak intensities of sample **A** decreased approximately by 90% and 70% for Phe 4 and Phe 94 peaks, respectively, but sample **C** was stable (just slight peak shifting was observed) after 17.4 h of aggregation.

Different stability (aggregation kinetics) was observed for the samples **B** and **D**. First of all, the intensity of the Phe 4 peak decreased almost twice and was very similar to the spectrum of sample **D** before aggregation (Figure 6B,D,F). This observation might indicate that a short (1.5 h) incubation of sample **B** at 37 ${}^{\circ }$C, 250 RPM induced S100A9 the same conformation or binding mode to α-syn as in the sample with calcium (as in sample **D**). This conformation of α-syn in sample **B** was stable also after 3.5 h at aggregation conditions (Figure 6B). Both peaks decreased after 12.7 h at a similar rate, but Phe 94 peak lost more intensity in a time period from 12 to 14.5 h (Figure 6F). The Phe 4 peak of sample **D** decreased in intensity and shifted more rapidly compared to Phe 94 (Figure 6D–F). Both peaks of sample **D** decreased and shifted almost the same after 12 h and later after 14.5 h did not change much. The midpoints (half time) of Phe 4 and Phe 94 intensity decrease (followed by exponential decay) of sample **D** were 2.9 h and 4.6 h, respectively, which suggests that somewhat different involvement in the aggregation process for protein regions occurs close to Phe 4 and Phe 94. By comparing the behavior of how Phe 4 and Phe 94 peaks decreased during aggregation, it was clear that the Phe 94 peak of sample **C** lost the intensity much more rapidly from approximately 12 to 14.5 h, while both peak intensities in the spectrum of sample **D** reached a plateau in this period of time and the intensity was higher (Figure 6E). The conclusions of these experiments would be that (1) calcium stabilizes α-syn; (2) S100A9 accelerates aggregation of α-syn; and (3) S100A9 in the presence of calcium leads to a different aggregation pathway than in the absence of calcium.

The amino acid residues of α-syn possibly involved in the first steps of aggregation or soluble oligomer formation were determined by solution 2D ^15^N–^1^H HSQC NMR spectroscopy. The peak intensity and chemical shift changes were analyzed for samples **A**–**D** after aggregation at 37 ${}^{\circ }C$, 250 RPM for 14 h. By comparing the intensity change in spectra of samples **A**–**D**, we observed that samples **B** and **D** (both containing α-syn and S100A9) showed larger changes compared to samples **A** and **C** (composed of α-syn without S100A9) (Figure 6G,I). The average decrease of peak intensity was 3%, 17%, 6% and 15% for samples **A**–**D**, respectively. The most affected region by the aggregation was the N-terminal part starting from the first assigned amino acid residue Val 3 and ending with Ser 9. This region showed not only large decrease of peak intensity but also high peak shifts for residues Val 3, Phe 4 and Met 5 (Figure 6H–J).

The largest intensity decrease was observed for the peaks assigned to the amino acid residues Ser 9, Gly 14 (overlapped peak clustered with Gly 106 and Gly 132), Asn 65, Gly 93 and Asn 103 in sample **A** (Figure 6H). The largest intensity decrease for sample **B** was measured for peaks in the region from Val 3 to Ser 9 (included Phe 4, Met 5, Lys 6, and Leu 8), Gly 14, Gly 36, Gly 41, Gly 51, the region from Ala 85, Gly 86, to Ser 87, Gly 93 and Asn 103 (Figure 6H). The most affected amino acid residues of sample **C** were Ser 9, Gly 14, Gly 36, Asn 65, Gly 93 and Asn 103 (Figure 6I). The peaks of sample **D**, which showed the largest decrease of intensity, were almost the same as for sample **C**: the region from Val 3 to Ser 9, Gly 14, Gly 36, the region from Ala 85 to Ile 88, Gly 93 and Asn 103 (Figure 6I). The C-terminal part starting approximately from residue Val 118 showed relatively large difference in intensity and chemical shift changes when sample **A** is compared with **B** (Figure 6G,H), showing that this region is sensitive to the presence of S100A9. This difference is smaller and starting just from Met 127 when sample **C** is compared with **D**, which indicates that S100A9 in the presence of calcium affects the α-syn C-terminal part to a lower extent (Figure 6I,J). The position of peaks in the ^15^N–^1^H HSQC spectrum did not change much for most peaks, but the peaks of Val 3, Phe 4, Met 5 and His 50 shifted significantly in all samples (Figure 6H,J). The shifts for peaks Val 3 and Phe 4 were higher for sample **D** compared to sample **C** (Figure 6J). The chemical shift change of Ser 9 was larger for the sample containing S100A9 (Figure 6H,J). The chemical shift changes of the peaks Tyr 33, Asp 119, Asp 121, Asp 122, Glu 126, Met 127, Ser 129, Asp 135 and Glu 137 were larger for sample **A** compared with the other samples (Figure 6H,J). This observation might show that the α-syn structure is more flexible without the presence of calcium or S100A9.

### 2.5. Aggregation of S100A9 by NMR Spectroscopy

The aggregation of ${}^{15}$N S100A9 resulted in a clear decrease of peak intensities in its ^15^N–^1^H HSQC spectrum after 3 h and 7 h at 37 ${}^{\circ }$C, 250 RPM. The study was done with two samples: (1) **S100A9**–${}^{15}$N S100A9 without α-syn and (2) **S100A9-syn**–${}^{15}$N S100A9 with the same molar concentration of α-syn. The average intensity decrease of all peaks was 21% after 3 h of aggregation for both samples, but it differed between samples after 7 h and was 30% and 60% for the samples **S100A9** and **S100A9-syn**, respectively (Figure 7A,C). Despite the total decrease of signal intensity, there were several separate amino acids or regions, which showed a distinctly larger (close to average plus two standard deviations) decrease of intensity, especially after 3 h of aggregation. The largest region that experienced the decrease of intensity was an intrinsically disordered C-terminal part starting from residue Leu 86 to almost the end of the protein sequence, excluding a few amino acid residues (Figure 7A). The short regions located at the N-terminal part (Gln 7, Glu 9, Glu 13) also showed an intensity decrease by 32% to 43%. The peaks of Phe 19, His 20, Glu 36, Ser 75, and Leu 82 displayed a larger reduction in signal intensity for sample **S100A9** compared to sample **S100A9-syn** when aggregation was performed for 3 h (Figure 7A). The opposite behavior was observed for residues Gln 7, Glu 9, Glu 13, Val 58, Ile 59, and Glu 60. The largest chemical shift changes after 3 h of aggregation were observed for Asp 30, Glu 60, His 61, Met 63 (for **S100A9-syn**), and His 105 (Figure 7B).

The most affected amino acid residues were almost the same for the sample **S100A9** after 7 h compared to 3 h of aggregation. This behavior was different for sample **S100A9-syn**, which showed a large reduction of intensity for almost all peaks up to Ser 90. We can emphasize a few of the most affected amino acid residues: Ser 6, Gln 46, Asn 47, Phe 48, Leu 49, Lys 50, Lys 51, Glu 52, Asn 53, Glu 56, Val 58 to Glu 60, Leu 86, Trp 88 and Ala 89. Chemical shift perturbations were also observed for these amino acid residues. The intensity of the peaks at the C-terminal intrinsically disordered part starting after Helix-4 was very similar for both samples except for a few peaks (Gly 96, Gly 102, His 103, Gly 112), showing that this part is flexible in the protein aggregate.

### 2.6. Fibril Morphology, Secondary Structure and Toxicity

The AFM images showed that amyloid fibrils were formed in the samples **A**–**D** (Figure 8A–D). The length of fibrils was similar in all samples ranging from 0.1 μm to 3 μm. A more objective parameter describing the morphology related to the protein structure in the fibrils is a cross-sectional height [54]. This parameter was higher for sample **A** (5.7 nm) compared with samples **B**, **D** (from 4.8 nm to 5.2 nm) (Figure 8E), showing that S100A9 or the presence of calcium induces a different structure of α-syn fibrils. This difference was statistically significant (*p* < 0.01) evaluating 50 fibrils of each sample by one-way ANOVA and post hoc Tukey test. The difference of cross-sectional height between sample **B** and **C** or **D** was lower but also significantly different (*p* < 0.01), showing that the presence of calcium induces a different structure of α-syn fibrils. The cross-sectional height was the same for samples **C** and **D**, indicating that S100A9 does not influence the fibril structure of α-syn in the presence of calcium.

We analyzed the profile of the amide band signals in FTIR spectra, which reflects the hydrogen bonding in the amyloid structure and thus is a qualitative analysis. The FTIR spectra of fibril samples **A**–**D** showed peaks in the region which is associated with hydrogen bonding in the β-sheet structure [55]. The spectrum of sample **A** and **C** showed maxima at 1626 cm^−1^ (Figure 8A). The main spectrum maximum position was shifted to 1628 cm^−1^ for samples **B** and **D**, which shows that the structure of samples **B** and **D** have weaker hydrogen bonding in a β-sheet structure. The spectrum of samples **B** and **D** had also shoulders at 1638 cm^−1^, which show the presence of different hydrogen bonding. The spectra coincide with ones from our previous work with α-syn and S100A9 [45].

Cytotoxic effects of the fibril samples **A**–**D** were investigated on SH-SY5Y human neuroblastoma cells by 3-(4,5-dimethylthiazol-2-yl)-2,5-diphenyltet-tetrazolium bromide (MTT) assay [56]. The fibrils of samples **A**–**C** at the fibril concentration of 40 μM reduced cell viability by 19% to 31% as compared to the control (Figure 8B). The toxic effect of sample **B** fibrils was observed to be milder: fibrils caused up to a 20% reduction in cell viability. The fibrils of sample **D** showed no toxic effect on cells as compared with the control.

The tendency of α-syn–S100A9 fibrils showing mitigated toxicity toward cells was also observed during experiments with fibrils, which were produced at 50 ${}^{\circ }$C (Appendix A
Figure A5). In this case, α-syn fibrils reduced cell viability by 13% to 28%, while α-syn–S100A9 did not decrease cell viability by more than 14%.

An experiment on lactate dehydrogenase (LDH) release [57] into cell medium was conducted to observe the influence of fibrils on the impairment of cell membrane and/or cell lysis. Fibrils of the sample **C** had an elevated damaging effect on the cells, but the other samples did not show LDH release. This tendency was very similar to the fibrils obtained at 50 ${}^{\circ }$C, where the highest toxicity was also for the sample **C** (Appendix A
Figure A5). Overall, tests of fibril toxicity toward cells showed that the co-aggregation of α-syn with S100A9 led to a reduced damaging effect on cells. Such a tendency was also observed in a previous study as well [29].

### 2.7. ${}^{13}$C, ${}^{15}$N α-syn Fibrils by FTIR and ssNMR

We conducted ssNMR studies for fibrillated α-syn and its complex with the S100A9 protein to obtain insights about the structural modifications of protein–protein interactions at the atomic level. First of all, we examined two ${}^{13}$C, ${}^{15}$N labeled α-syn samples (α-syn fibrils and α-syn fibrils aggregated in the presence of S100A9) by FTIR spectroscopy, which were prepared for ssNMR experiments. The spectra for both samples looked very similar, with the main maxima at 1583 cm^−1^ (Figure 9A) and corresponding minima in their second derivatives at the same position (Figure 9B), suggesting a predominant beta-sheet structure (as carbonyl groups were ${}^{13}$C-labeled, the wavenumbers expected to be approximately 35–37 cm${}^{-1}$ lower [58,59]).

The minor difference which could be observed was a small shoulder visible in the second derivative at 1593 cm^−1^ for the α-syn sample, which was aggregated in the presence of S100A9. It may suggest a small population of different hydrogen bonds involved in beta-sheets.

The fingerprint 2D NCa ssNMR spectra for both samples clearly exhibited chemical shift perturbation and also differences in peak intensities (Figure 9C). These changes can be related to conformational modifications induced by S100A9 binding to α-syn. The measured ${}^{13}$C line-width (around 120 Hz) for both samples indicated a high degree of structural homogeneity in amyloid fibrils. The chemical shift values for two of our samples compared to other ssNMR studies of α-syn [17,51] were different, revealing that the α-syn fibrils in our sample were structurally different. Further on, we will conduct ssNMR assignment studies to identify exact S100A9 binding sites in α-syn and distance restraints for the structure calculations.

## 3. Discussion

Using ${}^{19}$F NMR spectroscopy, we showed that the N-terminal part of α-syn interacts with S100A9. We confirmed this and obtained more details about specific interacting regions both from α-syn and S100A9 sides using 2D ^15^N–^1^H HSQC NMR spectroscopy. The N-terminal part of α-syn binds to the interface of S100A9, which is formed of Helix 1 and Helix 4 including residues 86–96. The same interface is occupied when S100A9 forms a homodimer or heterodimer with other S100 family (S100A8, S10012) proteins or receptor RANGE V [60]. The amino acid resides Leu 40 and Phe 48 of S100A9, which showed a high-intensity decrease after α-syn binding, are in the conserved hydrophobic patch found in many S100 family proteins, and it was shown to bind detergents, peptides, and proteins [36]. The C-terminal region from Arg 85 to Met 94 was reported to be involved in the S100A9 interaction with S100A8 when the dimer is formed. The amino acid residues in this region (residues 86–94) showed a large decrease in intensity in our study, indicating that this region is very important also to interact with α-syn.

To our knowledge, this is the first study of 4F-Phe labeled α-syn. We demonstrated that this probe is very sensitive to conformational changes of α-syn when it interacts with S100A9. This method allowed us to determine that calcium ions enhanced the interaction of the α-syn N-terminal part with S100A9. The calcium and S100A9 bound conformation of α-syn was also observed for the sample without calcium but with α-syn and S100A9 was incubated at 37 ${}^{\circ }$C, 250 RPM for 1.5 h. This conformation was stable for several hours at the mentioned incubation condition. The other studies showed that 4F-Tyr labeled α-syn can be used to analyze the binding of α-syn with membrane surfaces [61,62], detergent vesicles and the aggregation process [63]. These studies demonstrated that the 4F-Tyr 39 signal in the NMR spectrum changed the intensity or position when α-syn interacted with SDS micelles, small unilamellar vesicles, or plasma membrane, which shows that the region around Tyr 39 is involved in the interaction (consequently, the transient structure formation at the N-terminal part). Using 2D ^15^N–^1^H HSQC NMR spectroscopy, we also determined that the region close to Tyr 39 is involved in the interaction with S100A9. The previous studies by NMR spectroscopy showed that α-syn can interact with S100A9 at several regions (including the N-terminal part), but a slightly larger intensity decrease in the ^15^N–^1^H HSQC spectrum was observed in the C-terminal part of α-syn, which suggests the stronger interaction in that part [29]. This study analyzed a sample that contained a much higher S100A9 concentration (1:32 molar ratio of α-syn:S100A9), less salt, and no calcium, which explains why the study identified the C-terminal part as a stronger interacting region. However, the mentioned study partially confirms our result that the interaction of α-syn N-terminal with S100A9 is weaker without the presence of calcium.

The MD simulations confirmed that the interacting amino acid residues determined by NMR spectroscopy were important in complex formation. The simulation showed that the α-helical structure at the N-terminal part of α-syn might be stable in a complex with S100A9. The structure formation in α-syn after interaction with S100A9 was observed in the previous study by CD spectroscopy [29], which also supports our modeling results.

Tracking the aggregation of α-syn by measuring the signal intensity of fibril-bound ThT revealed how S100A9 and calcium affect this process. In both cases where S100A9 was present, the fluorescence intensity was significantly lower than the control, and the data dispersion was relatively low. This falls in line with our previously shown data [45], where S100A9 stabilized the formation of one α-syn fibril population. When α-syn was aggregated in the presence of calcium, there was a very clear sample distribution into three signal intensity populations (low, intermediate, and high), suggesting the formation of fibrils with distinct dye-binding modes. In all four cases, the aggregation itself proceeded with a high level of stochasticity, with a notable difference in the lack of low $t_{50}$ value populations and higher average $t_{50}$ values for samples with S100A9.

NMR data (both ${}^{19}$ F and ^15^N–^1^H HSQC) showed that α-syn aggregates faster when S100A9 is present. This is in contrast to results obtained by ThT assay. One of the reasons might be that we observed the first aggregation events (small oligomer formation) by NMR spectroscopy, which occurred before the start of transition observed by ThT fluorescence assay. The other factor affecting aggregation kinetics could be that α-syn samples in the ThT assay were incubated at aggregation conditions without any brake. Using NMR spectroscopy, we had to make long incubation periods at lower temperature to record NMR spectra (40 min at 15 ${}^{\circ }$C to record 1D ${}^{19}$F spectra for 4F-Phe α-syn samples and 90 min at 10 ${}^{\circ }$C to record ^15^N–^1^H HSQC spectra for ${}^{15}$N α-syn samples), which made the process of aggregation different from what was done by ThT assay.

${}^{19}$F and 2D NMR spectroscopy showed that the N-terminal part of α-syn aggregates faster than the C-terminal part when the sample contains S100A9. This sample showed a slightly different profile of intensity decrease in fluorine spectrum compared to the α-syn sample with S100A9 but without calcium. This observation suggests that S100A9 and calcium might initiate different aggregation pathways. In addition, 2D ^15^N–^1^H HSQC NMR experiments showed that the presence of S100A9 affected also the aggregation of the C-terminal part of α-syn. The C-terminal part of S100A9 also showed large changes during the first few hours of aggregation independent on α-syn, and later, this part stayed stable, but the rest of the S100A9 sequence aggregated further only if α-syn was present. Based on these observations, we can speculate that when S100A9 binds to α-syn, for the first few hours, the dimerization or tetramerization occurs of S100A9, and afterward, the larger oligomers are formed, which depends on both S100A9 and α-syn proteins presence.

The general fibril morphology was similar for all samples determined by AFM, but the statistical analysis of the fibril cross-sectional height showed the higher difference between fibrils of sample **A** compared to the other samples. This observation indicates that the presence of S100A9 or/and calcium can change the structure of fibrils. FTIR spectroscopy showed that the secondary structure of α-syn fibrils does not depend much on the calcium, but the differences are higher if S100A9 is present. The solid-state NMR data confirmed our present and previous studies [45], indicating that the structure of α-syn fibrils is different when α-syn is aggregating in the presence of S100A9. Moreover, this different α-syn fibril structure results in a lower toxicity of cell culture. We can speculate that S100A9 interaction with α-syn can have a protective function to reduce the toxicity of fibrils or oligomers. There was also an observable difference in how α-syn fibrils obtained in the presence of calcium can affect cell plasma membrane permeability. Combining the used methods, we can conclude that calcium and S100A9 affect the final structure of α-syn.

## 4. Materials and Methods

### 4.1. Protein Production

The expression and purification of α-syn was performed as described previously [45]. In short, α-synwas expressed in *E.coli* BL21(DE) one-star strain at 30 ${}^{\circ }$C for 12 h. Purification was done using heat denaturation, pelleting by ammonium sulfate, and ion exchange (chromatogram and SDS-PAGE gel of pure α-syn are shown in the appendix, Figure A6B,E).

4F-Phe labeled α-syn was expressed in an auxotrophic E.coli cell line—BL39(DE3) by growing it in the selectively labeled medium described previously [64]. The BL39(DE3) was transformed with pRK172 plasmid containing human α-syn cDNA and grown on an LB agar plate with 0.1 g/L ampicillin overnight. The selected colony inoculated in 5 mL LB medium with ampicillin and grown at 37 ${}^{\circ }$C, 200 RPM for several hours until the culture reached the optical density (OD) at 600 nm of 0.6. This culture was added to a fresh 0.5 L LB medium with ampicillin and was grown again the same way as has just been described. When the OD at 600 nm reached 0.5, the culture was harvested by centrifugation (5 min, 4000× *g*) and resuspended into selectively labeled medium, containing 15 mg/L 4-fluoro-DL-phenylalanine (Fluorochem Ltd, UK). The culture was growing at 37 ${}^{\circ }$C, 200 RPM. When OD at 600 nm reached 0.8, the protein expression was induced by 1 mM IPTG and the culture was grown for 16 h at 20 ${}^{\circ }$C, 220 RPM. The harvested culture was used to purify 4F-Phe α-syn as described for not labeled α-syn.

The construct to express S100A9 was used as described previously [65]. The expression and preparation of cell-lysate of S100A9 were done the same as described for α-syn, except for the lysis buffer, which was without NaCl. The supernatant of cell-lysate was saturated with 70% ammonium sulfate solution on ice, pellets were separated by centrifugation, the supernatant was dialysed against 20 mM Tris-HCl buffer, 500 μM EDTA, 500 μM dithiothreitol (DTT, Fisher bioreagents, Canada), pH 8.0 for 3 h and loaded on a Q Sepharose FF (GE Healthcare, Sweden) a 20 mL column. S100A9 was eluted at approximately 250 mM NaCl concentration of the applied linear gradient. The purest fractions containing S100A9 were concentrated using spin filters (Amycon, 10 kDa MWCO) and purified by gel filtration using 20 mL Superdex 75 (GE Healthcare, Sweden) column and 200 mM of ammonium bicarbonate as elution buffer (Appendix A
Figure A6A). The pure (purity approximately 90% estimated from SDS-PAGE gel) S100A9 were concentrated up to 0.8 mM, freeze-dried and kept at −80 ${}^{\circ }$C.

${}^{15}$N labeled α-syn and S100A9 were separately expressed using *E. coli* BL21(DE) one star strain, growing it in M9 nutrition minimal medium (42.2 mM (6 g/L) Na_2_HPO_4_, 22 mM (3 g/L) KH_2_PO_4_, 2 mM MgSO_4_, 0.10 mM CaCl_2_ and a mixture of trace metals) with 1 g/L of ^1^5N labeled ammonium chloride, 4 g/L of D-glucose and 1 mg/L ampicilin. The culture was grown up to OD at 600 nm, reached 0.8, and the protein expression was induced with 1 mM IPTG. The culture was grown for 16 h at 20 ${}^{\circ }$C, and the protein purified as it was described for unlabeled α-syn and S100A9.

### 4.2. Mutagenesis to Obtain F4Y α-syn

The mutagenesis of α-syn was performed by introducing F4Y mutation using site-directed mutagenesis based on one PCR reaction, as previously reported [66,67]. Plasmid pRK172 containing human α-syn gene was used as template DNA and was added to the mutagenesis mixture together with 5${}^{\prime }$ phosphorylated primers: 5${}^{\prime }$-tacatgaaaggactttcaaagg and 5${}^{\prime }$-tacatccatatgtatatctcc. PCR was executed using Phusion DNA Polymerase (Thermo Scientific, Vilnius, Lithuania), which is one of the most accurate thermostable polymerases available, using the following guidelines: initial denaturation in 98 ${}^{\circ }$C for 2 min as the first step, denaturation in 98 ${}^{\circ }$C for 10 s, primer annealing in 57 ${}^{\circ }$C for 30 s and extension in 72 ${}^{\circ }$C for 4 min as the second step and final extension in 72 ${}^{\circ }$C for 10 min as the third step. The PCR reaction was performed by repeating the second step for 30 cycles.

The linear double-stranded DNA after the PCR reaction was purified from the PCR mixture by the gel extraction method using a GeneJet Gel Extraction Kit (Thermo Scientific, Lithuania). The PCR product was ligated with T4 DNA ligase (Thermo Scientific, Lithuania) by executing the ligation reaction at room temperature (20 ${}^{\circ }$C) for 1 h, which was followed by ligation mixture incubation at 25 ${}^{\circ }$C for 20 min. Due to the use of phosphorylated primers, there was no need to perform a DNA phosphorylation reaction before the PCR product ligation. The circular plasmid pRK172-F4Y was amplified by performing transformation to *E. coli* XL1 Blue cells, which was followed by E.coli culture growth in LB medium and ampicillin containing sterile flasks. The amplified plasmid was purified from bacteria culture using a GeneJET Plasmid Miniprep Kit (Thermo Scientific, Lithuania), and the mutation was tested by sequencing the purified plasmid. The mutant protein was expressed and purified the same way as described for 4F-Phe α-syn.

### 4.3. α-syn Binding with S100A9 by NMR Spectroscopy

All NMR spectra were recorded using a 600 MHz Bruker Avance Neo spectrometer equipped with a cryogenic probe. The 1D ${}^{19}$F NMR spectra of 4F-Phe labeled α-syn were recorded in 20 min using 256 scans. The protein solution filled in a 5 mm NMR tube contained: 50 μM 4F-Phe α-syn, 0 μM, 35 μM, 50 μM, 70 μM, 125 μM, 250 μM, 500 μM S100A9, 25 mM HEPES, 100 mM NaCl, 5 mM DTT, 0 mM or 5 mMCaCl_2_, 5% D_2_O, 0.02% sodium azide, pH 7.4. Two-dimensional (2D) ^15^N–^1^H HSQC spectra of α-syn were recorded at 10 ${}^{\circ }C$ or 15 ${}^{\circ }C$ using 256 increments in the indirect dimension and 8 scans. Protein solution composed of 50 μM${}^{15}$N α-syn, 0 μM or 500 μM S100A9, 25 mM HEPES, 100 mM NaCl, 5 mM DTT, 5 mM CaCl_2_, 5% D_2_O, 0.02% sodium azide, pH 7.4.

Two-dimensional (2D) ^15^N–^1^H HSQC spectra of S100A9 were recorded at 25 ${}^{\circ }$C or 30 ${}^{\circ }$C using 128 increments in the indirect dimension and 64 scans. Protein solution 100 μM${}^{15}$N S100A9, 0 μM, 125 μM or 500 μMα-syn, 25 mM HEPES, 100 mM NaCl, 5 mM DTT, 5 mM CaCl_2_, 5% D_2_O, 0.02% sodium azide, pH 7.4.

### 4.4. MD Simulations

The starting conformation of α-syn-S100A9 complex was prepared by aligning S100A9 structure 5I8N and 50 first residues of α-syn structure 1XQ8 with calmodulin and α-syn structure 2M55. Furthermore, protein structures were prepared using Maestro [68] Protein Preparation Wizard [69] by adding missing side chains using Prime [70], adjusting side-chain protonation states at pH 7.4, and minimizing heavy atoms with convergence up to 0.30 *Å*. The MD simulation systems of ligand–enzyme complexes were prepared by placing the molecules in a dodecahedral box with at least 1.5 nm distance to the box walls. The TIP3P water model was used to solvate the complex. Sodium and chloride ions were added to neutralize the systems and reach 150 mM salt concentration. All simulations were carried out with the AMBER99SB-ILDN force field [71,72] for the protein and the TIP3P water model [73] for the explicit solvent. The prepared systems were relaxed through an energy minimization, using the steepest descent algorithm with a tolerance of 100 kJ/(mol × nm). After minimization, systems were equilibrated in the NVT and then NPT ensembles for 5 ns. The MD (leapfrog) integration scheme with an integration time step of 2 fs was employed for equilibration and production runs. The particle mesh Ewald (PME) approach was used to calculate long-range electrostatic interactions with a cut-off of 0.8 nm. Both Lennard–Jones and Coulomb interactions were explicitly calculated up to 0.8 nm. The LINCS algorithm [74] was applied at each step to preserve the hydrogen bond lengths. NPT equilibration was performed employing a Berendsen barostat [75] with a coupling constant of 2 ps and reference pressure of 1 . Velocity-rescale thermostat [76] with a coupling constant of 2 ps and reference temperature 298.0 K was used for equilibration and production simulations. The production runs were performed in the NPT ensemble for 100 ns. Then, 20 parallel production runs were performed. System coordinates were saved every 10 ps, and a total of 2,000,000 frames were generated for further analysis. The potential energy minimization and MD simulations were carried out with the software package Gromacs 2021 [77,78].

Analysis was performed on a 1 long concatenated trajectory, where the first 50 ns of each parallel simulation was discarded. The clustering of sampled protein complexes was performed on 20,000 frames, using the GROMOS clustering algorithm [53] with the root mean squared deviation (RMSD) as a measure of similarity between matrices. The RMSD cut-off used was 0.3 nm. Contact maps for the 10 most populated clusters were computed by considering interactions within a 1 nm cut-off. Contact map distances were discretized in 10 levels. Trajectory clustering and contact map generation were performed using the software package Gromacs 2021. Trajectories were analyzed using VMD software [79], and figures were prepared using Pymol software.

### 4.5. ThT Assay

Protein aggregation kinetics by ThT assay [80] were performed in a 96-well Corning non-binding half-area plate with one glass bead (diameter 3 mm) in each well. The plate, filled by 100 of protein solution in each well, was shaking constantly between readings (every 10 min) using 300 RPM orbital agitation in a CLARIOstar Plus (BMG Labtech, Ortenberg, Germany) plate reader at 37 ${}^{\circ }$C for 35 h. Sample fluorescence intensity was measured using excitation and emission wavelengths of 440 nm and 480 nm, respectively. All protein samples contained 100 μMα-syn, 0 μM or 100 μM S100A9, 50 μM ThT, 25 mM HEPES buffer, 0.1 M NaCl, 0.02 NaN_3_, 1 mM ditiotreitole (DTT), 0 mM or 1 mMCaCl_2_, and pH 7.4. The $t_{50}$ was determined by fitting the sigmoidal curve to aggregation kinetic data.

### 4.6. α-syn and S100A9 Aggregation by NMR Spectroscopy

The aggregation studies of 4F-Phe labeled α-syn, ${}^{15}$N labeled α-syn and ${}^{15}$N S100A9 were performed in a 96-well plate by one sample spread into 8 wells; each contained 100 of protein solution. One glass bead (3 mm in diameter) was placed in each well. The protein solution was composed of: 0 μM or 100 μMα-syn (4F-Phe, ${}^{15}$N labeled or not specifically labeled), 0 μM or 100 μM S100A9 (${}^{15}$N or not specifically labeled), 0 μM or 1 mM CaCl_2_, 25 mM HEPES, 100 mM NaCl, 1 mM DTT, 5 % D_2_O, 0.02 % NaN_3_, pH 7.4. The plate was shaken at 250 RPM at 37 ${}^{\circ }$C between measurements. Before each measurement started, the particular sample was taken from all wells and added to a 5 mm NMR tube. It took from 30 to 40 min for recording the 1D ${}^{19}$F spectrum (256 scans) and 1.5 h for the 2D spectrum (256 increments in indirect dimension, 8 scans) of α-syn at 10 ${}^{\circ }$C. Four spectra were recorded in between the first and the last spectra (21.5 h of experiment included: (1) 14.5 h incubation time at 37 ${}^{\circ }$C, (2) 7 h for measurements at 10 ${}^{\circ }$C). Recording 2D ^15^N–^1^H HSQC of S100A9 took 2.5 h using 128 increments and 64 scans. Data were analyzed using Topspin 4.1.3 and CcpNMR 2.4.1 analysis [81] software. The chemical shift perturbation was calculated as described previously [82]:(1)$$CSP=\sqrt{{\left(\delta^{H}-\delta_{0}^{H}\right)}^{2}+0.01{\left(\delta^{N}-\delta_{0}^{N}\right)}^{2}},$$
where $\delta^{H}$ and $\delta^{N}$ are chemical shifts of H and N after some change of protein solution composition (in the binding experiments) or some time passed of aggregation process. $\delta_{0}^{H}$ and $\delta_{0}^{N}$ are the chemical shifts of H and N for the starting solution (in the case of binding experiment) or time point (in the case of protein aggregation).

### 4.7. Atomic Force Microscopy (AFM)

Fibril morphology was analyzed by AFM (DimensionIcon, Bruker) the same way as described previously [45]. In short, the fibril samples were diluted 20 times with water and added on the mica surface modified with APTES. The washed and dried surface was scanned by AFM to record 1024 × 1024 pixels per 10 μm images. The analysis was done using Gwyddion software [83]. The cross-sectional peaks were fitted by Gaussian function to determine the fibril heights. The statistical analysis was done by one-way ANOVA with the post hoc Tukey test implemented in QtiPlot software.

### 4.8. FTIR Spectroscopy

Multiple samples were combined into 2 groups based on ThT fluorescence intensity. The samples were reseeded mixing 20% to 60% of fibrils with fresh protein solution and incubated at 37 ${}^{\circ }$C for 24 h. Each sample were combined and centrifuged at 12,500 RPM for 10 min, the supernatant was removed and the fibril pellet was resuspended into 200 D_2_O (with 400 mM NaCl to improve sedimentation) [84]. After repeating the centrifugation and resuspension procedure three times, the fibril pellet was resuspended into a final volume of 60 D_2_O. For each sample, 256 interferograms with 2 cm^−1^ resolution were collected using a Bruker Invenio S spectrometer equipped with a liquid-nitrogen-cooled mercury–cadmium–telluride detector at room temperature in a dry-air flow-through chamber. A D_2_O spectrum was subtracted from each sample spectrum, after which the spectra were baseline corrected in the region at 1595 cm^−1^ to 1700 cm^−1^ and normalized. All data processing was done using GRAMS software.

### 4.9. ssNMR Spectroscopy

The ssNMR spectra were acquired at 12 kHz MAS using an HCN Bruker e-free probe in a 18.8 T Bruker magnet. The sample temperature was set to 4 ${}^{\circ }$C. The 2D NCa spectra for both samples were acquired with t1 of 9.0 ms, t2 of 17.2 ms, relaxation delay of 2.5 s and 204 scans, leading to a total experimental time of 12.4 h. Both dimensions for two spectra were processed with a sine bell shift of π/3 using Topspin 3.6.1 and analyzed with the CcpNMR2.4.1 program [81].

### 4.10. Cell Culture, Cell Viability and Lactate Dehydrogenase (LDH) Release Tests

The human neuroblastoma SH-SY5Y cell line was obtained from ATCC and maintained at 37 ${}^{\circ }$C in a humidified incubator (5% CO${}_{2}$/95% air) in Dulbecco’s Modified Eagle Medium (DMEM) (Gibco, Grand Island, NY, USA), containing 10% fetal bovine serum (FBS) (Sigma-Aldrich, St. Louis, MO, USA) and 1% penicillin and streptomycin (Gibco, Grand Island, NY, USA). For cell viability and LDH release assays, SH-SY5Y cells were seeded in a plate of 96 wells (TPP, Trasadingen, Switzerland) one day before the fibrils addition. Prior to adding on cells, fibrils were centrifuged (10,000 RPM, 10 min) and washed with PBS. During the MTT test, fibrils were suspended in 100 fully supplemented DMEM medium. The medium inside the 96-well plate was aspirated and changed with one containing fibrils. After 48 h of incubation, 10 of MTT reagent (Invitrogen, Waltham, MA, USA) dissolved in PBS (5 mg/mL) was added to each well and left to incubate for another 2 h. Formazan crystals were dissolved in 100 of 10% SDS with 0.01 N HCl solution, and the absorbance of each well was measured at the wavelengths of 570 nm and 690 nm (reference wavelength) by a ClarioStar Plus plate reader (BMG Labtech, Ortenberg, Germany). During LDH release monitoring, the test procedure was carried out according to the cytotoxicity detection kit manufacturer (Roche Applied Science, Germany) protocol. Prior to adding fibrils, the medium of each well was changed with Advanced DMEM (Gibco, Grand Island, NY, USA). Fibrils were suspended in 100 of Advanced DMEM and applied to cells. After 24 h of incubation, 120 of the medium was aspirated from each well, centrifuged and 100 of supernatant was transferred into a 96-well plate (Corning, Tewksbury, MA, USA). Then, 30 min after the addition of LDH reagent to each well, absorbance was measured at the frequencies of 492 nm and 600 nm (reference wavelength) using a ClarioStar Plus plate reader. All experiments were performed in triplicate. Values are given as a mean and standard deviation. Student’s *t*-test was used to evaluate the statistical significance between the groups with a probability of * *p* < 0.05, ** *p* < 0.01, and *** *p* < 0.001.

## 5. Conclusions

We can conclude based on NMR data that the N-terminal part of α-syn interacts with S100A9. MD simulations support this conclusion and allow us to select the possible complex structures. This study showed also that it is possible to study the interaction and aggregation of 4F-Phe labeled α-syn, which allows distinguishing conformational changes and the stability of α-syn N-terminal and NAC parts separately. Two-dimensional (2D) ^15^N–^1^H HSQC NMR experiments can be applied to study α-syn and S100A9 interaction and aggregation. The NMR experiments allow us to conclude that (1) calcium stabilizes α-syn; (2) S100A9 accelerates the aggregation of α-syn; and (3) S100A9 in the presence of calcium leads to a different aggregation pathway than in the absence of calcium. Finally, we can conclude that calcium and S100A9 influence the pathway of α-syn fibril formation and a final structure.

## Figures and Tables

**Figure 1 ijms-23-06781-f001:**
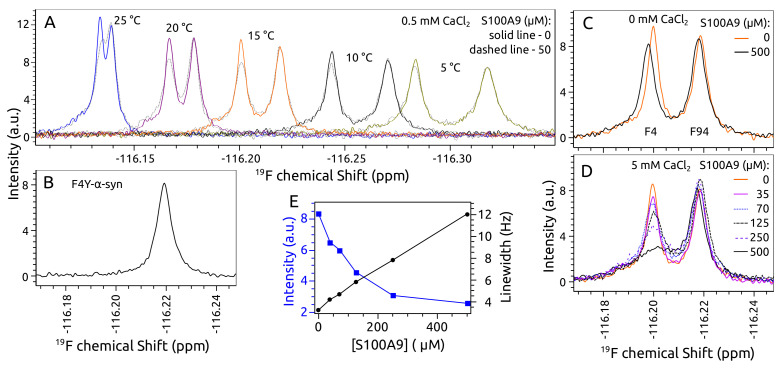
One-dimensional (1D) ${}^{19}$F NMR spectra. of 4-fluoro-Phe 4 (F4 peak) and 4-fluoro-Phe 94 (F94 peak) labeled α-syn: (**A**) spectra at different temperatures from 25 to 5 ${}^{\circ }$C in absence (solid line) and presence (dotted line) of 50 μM S100A9, (**B**) F4Y α-syn mutant, (**C**) spectra in absence (orange spectrum) and presence (black) of 500 μM S100A9 without addition of calcium chloride, (**D**) spectra in the presence of different concentrations of S100A9 and 5 mMCaCl_2_. (**E**) The line width and intensity change of α-syn F4 peak at different concentrations of S100A9. Protein solution: 50 μMα-syn, 0 μM, 35 μM, 50 μM, 70 μM, 125 μM, 250 μM, 500 μM S100A9, 25 mM HEPES, 100 mM NaCl, 5 mM DTT, 0 mM or 5 mMCaCl_2_, 5% D_2_O, 0.02% sodium azide, pH 7.40. Spectra in panels B–D were recorded at 15 ${}^{\circ }$C.

**Figure 2 ijms-23-06781-f002:**
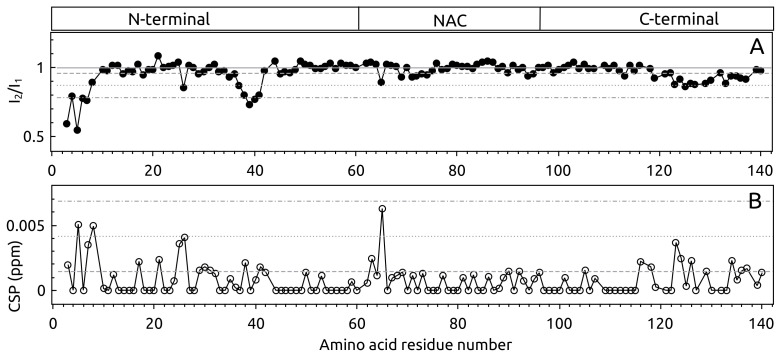
(**A**) Intensity and (**B**) chemical shift perturbation (CSP) of α-syn after addition of S100A9 at 10 ${}^{\circ }C$. Intensity change expressed as a ratio of α-syn intensities in the presence (I${}_{2}$) and the absence (I${}_{1}$) of S100A9. Horizontal gray lines: solid— starting point (no change of intensity), dashed—the average, dotted—average including standard deviation (STD), and dot dash—average including two STD of the signal intensity or chemical shift change. The structure segments of α-syn indicated above: N-terminal, Non-Amyloid beta Component (NAC) and C-terminal. Protein solution: 50 μM${}^{15}$N α-syn, 0 μM or 500 μM S100A9, 25 mM HEPES, 100 mM NaCl, 5 mM DTT, 5 mM CaCl_2_, 5% D_2_O, 0.02% sodium azide, pH 7.4.

**Figure 3 ijms-23-06781-f003:**
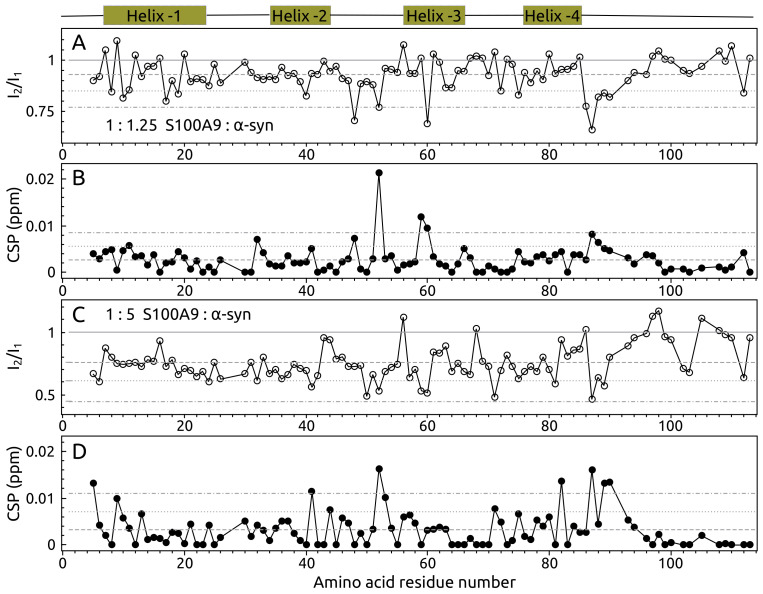
(**A**,**C**) Intensity change and (**B**,**C**) chemical shift perturbation (CSP) of S100A9 after addition of α-syn (molar ratio (**A**,**B**) 1:1.25 and (**C**,**D**) 1:5 of S100A9:α-syn). Intensity change was expressed as the ratio of S100A9 peak intensities in the presence (I${}_{2}$) or the absence (I${}_{1}$) of α-syn. Horizontal gray lines: solid, dashed, dotted and dash dot correspond to starting line, when α-syn has no influence, the average, average including standard deviation (STD) and two STD of signal intensity or chemical shift change. Protein solution: 100 μM${}^{15}$N S100A9, 0 μM, 125 μM or 500 μMα-syn, 25 mM HEPES, 100 mM NaCl, 5 mM DTT, 5 mM CaCl_2_, 5% D_2_O, 0.02% sodium azide, pH 7.40.

**Figure 4 ijms-23-06781-f004:**
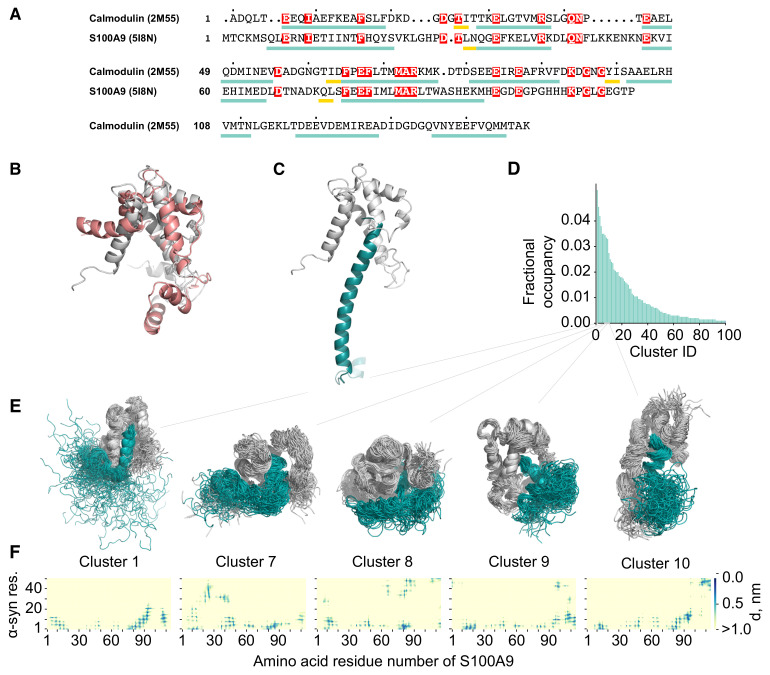
(**A**) S100A9 and calmodulin sequence and secondary structure alignment. Teal rectangles represent α-helixes; yellow rectangles represent β-sheets. Identical residues are shaded in red. (**B**) S100A9 and calmodulin structure superposition. Pink cartoon—calmodulin (PDB ID: 2M55); white cartoon—S100A9 (PDB ID: 5I8N). (**C**) α-syn-S100A9 complex used as a starting model in molecular dynamics simulations (constructed using 2M55 structure as a template). White cartoon—S100A9; teal cartoon—α-syn residues 1–50. (**D**) Bar plot showing fractional cluster occupancy. GROMOS clustering was performed considering both proteins and using a cut-off value of 0.3 nm. Fractional cluster occupancy is calculated on 20,000 frames. (**E**) Structures of selected clusters and (**F**) their inter-residue contact maps. Gray cartoon—S100A9; teal—α-syn.

**Figure 5 ijms-23-06781-f005:**
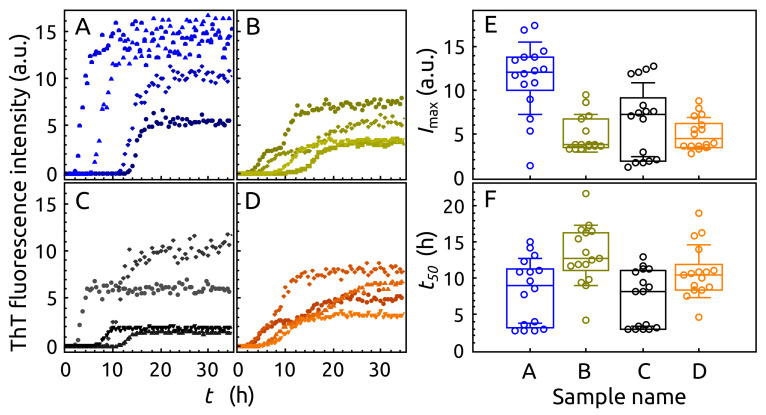
(**A**–**D**) The aggregation kinetic curves (4 selected of each sample, which represent stochastic process of 16 repeats) of α-syn (not labeled). (**E**) The largest value of ThT fluorescence intensity ($I_{max}$) and (**F**) $t_{50}$ values of samples **A**–**D**. The interquartile region (shown by the box), standard deviation (whiskers) and a median (horizontal line inside the box) were calculated from 16 curves of each sample. The protein samples: (**A**) α-syn, (**B**) α-syn with S100A9, (**C**) α-syn with CaCl_2_, (**D**) α-syn with S100A9 and CaCl_2_. Protein solution 100 μMα-syn, 0 μM or 100 μM S100A9, 25 mM HEPES, 100 mM NaCl, 0.02% sodium azide, 50 μM ThT, 1 mM DTT, 0 mM or 1 mMCaCl_2_, pH 7.4. Aggregation at 37 ${}^{\circ }$C, 300 RPM.

**Figure 6 ijms-23-06781-f006:**
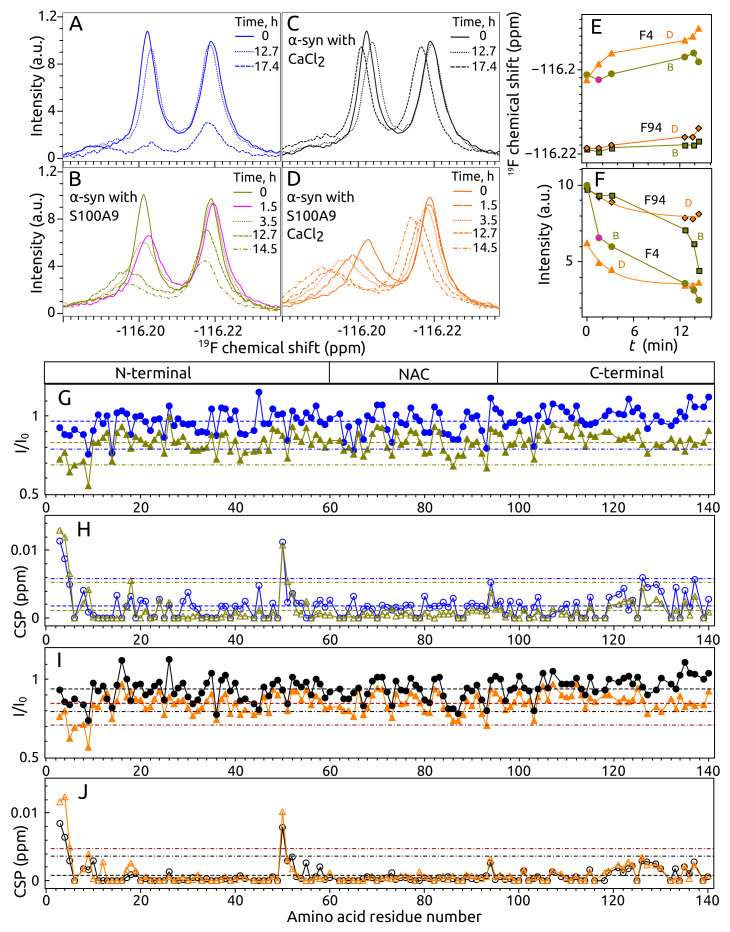
(**A**–**D**) ${}^{19}$F NMR spectra of α-syn at different conditions and time points of aggregation: (**A**) α-syn without CaCl_2_, (**B**) α-syn with CaCl_2_, (**C**) α-syn with S100A9 without CaCl_2_, (**D**) α-syn with S100A9 and CaCl_2_. (**E**) Chemical shift and (**F**) intensity of α-syn F4 (dark yellow circles or one pink circle—sample from panel **C**, orange triangles—**D**) and F94 (dark yellow squares—sample from panel C, orange diamonds—D) peaks at different time points during aggregation. (**G**,**I**) Intensity and (**H**,**J**) chemical shift change of ${}^{15}$N α-syn peaks in ^15^N–^1^H HSQC spectra after incubation at 37 ${}^{\circ }C$, 250 RPM for 14 h. (**G**,**H**) ${}^{15}$N α-syn solution without CaCl_2_, (**I**,**J**) ${}^{15}$N α-syn—with CaCl_2_. Circle points— α-syn, triangles ${}^{15}$N α-syn in a presence of S100A9. The dashed line is the mean and dash–dot line—two standard deviations (color coded lines and data points). All samples contained 100 μMα-syn, 0 μM or 100 μM S100A9, 0 μM or 1 mM CaCl_2_, 25 mM HEPES, 100 mM NaCl, 1 mM DTT, 5% D_2_O, 0.02% NaN_3_, pH 7.4. (**A**,**C**) Intensity and (**B**,**D**) chemical shift change of ${}^{15}$N α-syn after incubation at 37 ${}^{\circ }$C, 250 RPM for 14 h.

**Figure 7 ijms-23-06781-f007:**
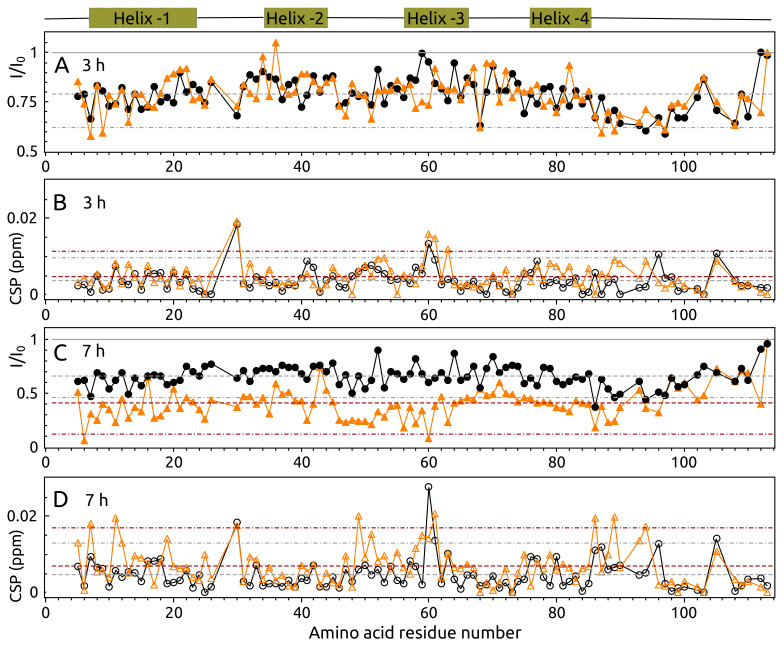
(**A**,**C**) Intensity and (**B**,**D**) chemical shift change of S100A9 peaks in ^15^N–^1^H HSQC spectra after incubation at 37 ${}^{\circ }$C, 250 RPM for 3 h (**A**,**B**) and 7 h (**C**,**D**) in the absence (black) or presence (orange) of α-syn. The schematic secondary structure of S100A9 is shown above. Solid gray line in (**A**,**C**) is the intensity of S100A9 before aggregation started (I${}_{0}$). I is the intensity after 1 h or 3 h of aggregation. The average of intensity or a chemical shift perturbation are shown by a dashed line, mean plus two standard deviations—dot dash line (gray color for sample **S100A9** and wine for **S100A9-α-syn**). Protein solution: 100 μM${}^{15}$N S100A9, 0 or 100 μMα-syn, 25 mM HEPES, 100 mM NaCl, 1 mM DTT, 1 mM CaCl_2_, 5% D_2_O, 0.02% sodium azide, pH 7.4.

**Figure 8 ijms-23-06781-f008:**
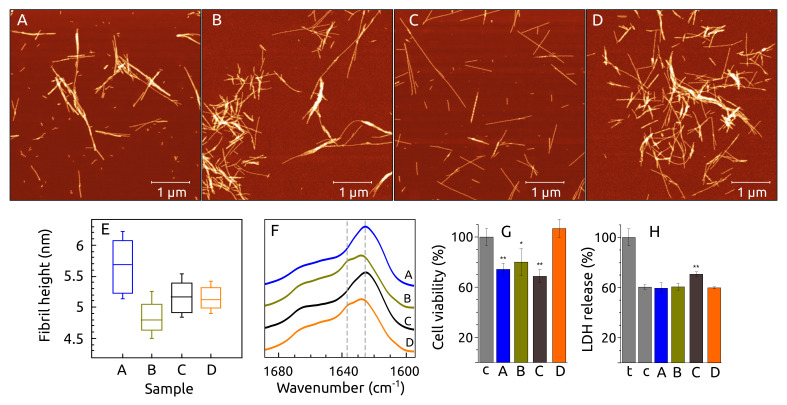
(**A**–**D**) AFM images, (**E**) fibril height, (**F**) FTIR spectra of fibril samples **A**–**B**. The influence of fibril for (**G**) SH-SY5Y cell viability and (**H**) LDH release (c—control, samples **A**–**D**, *t*–cells affected by detergent Triton X-100). Probabilities * *p* < 0.05, ** *p* < 0.01 show statistical significant differences between sample and control by Student’s *t*-test. Aggregation conditions: 100 μMα-syn, 0 μM or 100 μM S100A9, 25 mM HEPES, 100 mM NaCl, 0.02 sodium azide, 50 μM ThT, 1 mM DTT, 0 mM or 1 mM CaCl_2_, pH 7.4.

**Figure 9 ijms-23-06781-f009:**
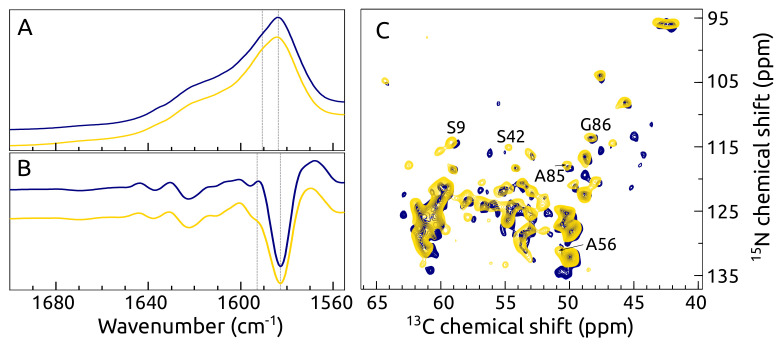
(**A**) FTIR spectra (**B**) the second derivative of FTIR spectra and (**C**) the overlay of 2D NCa ssNMR spectra of two ${}^{13}$C, ${}^{15}$N labeled α-syn samples: navy—α-syn fibrils, yellow—α-syn aggregated in a presence of S100A9. Protein solution for aggregation: 70 μMα-syn, 0 μM or 70 μM S100A9, PBS, 0.02 NaN_3_, pH 7.4.

## Data Availability

Not applicable.

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
