# Peer review of "Interactions between S100A9 and Alpha-Synuclein: Insight from NMR Spectroscopy"

_ijms, 2022, doi:10.3390/ijms23126781_

Round 1

Reviewer 1 Report

The article “Interactions between S100A9 and alpha-synuclein: insight from NMR spectroscopy” by Zigmantas Toleikis et al aims to document the cross-interaction of S100A9 and alpha-synuclein and understand how it influences the aggregation process. It is a timely article considering the demand for protein aggregation modifiers and their role in therapeutic intervention to neurodegenerative diseases. The authors have used the solution 19F and 2D 15N NMR spectroscopy to study the interaction. Authors reported that the 4-fluorophenylalanine label in alpha-synuclein is a sensitive probe to study interaction and aggregation using F-NMR spectroscopy. I am enthusiastic about this article and supportive of its publication. I only offer some minor suggestions to improve readability and enhance the message of the paper (adopting them is optional).

Minor issues: 

  1. Reference format is not same for all references

  2. FTIR data is not explained in detail. Looks like only qualitative analysis.

  3. Authors should discuss potential effects of introducing 4F-Phe if any.

Author Response

Our response is shown in blue.

Reviewer 1

Comments and Suggestions for Authors

The article "Interactions between S100A9 and alpha-synuclein: insight from NMR spectroscopy" by Zigmantas Toleikis et al. aims to document the cross-interaction of S100A9 and alpha-synuclein and understand how it influences the aggregation process. It is a timely article considering the demand for protein aggregation modifiers and their role in therapeutic intervention to neurodegenerative diseases. The authors have used the solution 19F and 2D 15N NMR spectroscopy to study the interaction. Authors reported that the 4-fluorophenylalanine label in alpha-synuclein is a sensitive probe to study interaction and aggregation using F-NMR spectroscopy. I am enthusiastic about this article and supportive of its publication. I only offer some minor suggestions to improve readability and enhance the message of the paper (adopting them is optional).

Minor issues:

Reference format is not same for all references

There are a several references of the software, which have different referencing style, as it is not a publication. The other references we checked and corrected.

FTIR data is not explained in detail. Looks like only qualitative analysis.

We analyze the profile of the amide band signals in FTIR spectra, which reflects the hydrogen bonding in the amyloid structure and thus is a qualitative analysis. The profiles look similar and the only noteworthy changes in our FTIR spectra upon co-aggregation with S100A9 was an increase in the weaker hydrogen bonding peak and a small shift of the main maximum towards lower wavenumbers (possibly higher content of stronger hydrogen bonding in the beta-sheets). We analyzed the influence of S100A9 to α-syn FTIR spectra in the previous study, which showed the same profile of spectra compared to this study (Toleikis et al. 2021).

Z. Toleikis, M. Ziaunys, L. Baranauskiene, V. Petrauskas, K. Jaudzems, and V. Smirnovas, “S100A9 Alters the Pathway of Alpha-Synuclein Amyloid Aggregation,” Int J Mol Sci, vol. 22, no. 15, p. 7972, Jul. 2021, doi: 10.3390/ijms22157972.

We added this reference in the manuscript.

Authors should discuss potential effects of introducing 4F-Phe if any.

Based on published data (Welte et al. 2020; Kitevski-LeBlanc et al. 2010), introduction of few fluorines should not change the structure and stability of proteins (we mentioned this in the introduction part). There might be some differences, but it did not change the binding or aggregation properties significantly as we confirmed it by analysis of 15N labeled α-syn. Both 4F-Phe and 15N labeled α-syn bound S100A9 in the same region. Regarding aggregation, S100A9 increased (sped up) aggregation of α-syn in both 4F-Phe and 15N α-syn samples, confirming that different labeling did not change the properties of α-syn.

H. Welte, T. Zhou, X. Mihajlenko, O. Mayans, and M. Kovermann, “What does fluorine do to a protein? Thermodynamic, and highly-resolved structural insights into fluorine-labelled variants of the cold shock protein,” Sci Rep, vol. 10, no. 1, p. 2640, Dec. 2020, doi: 10.1038/s41598-020-59446-w.

J. L. Kitevski-LeBlanc, F. Evanics, and R. Scott Prosser, “Optimizing 19F NMR protein spectroscopy by fractional biosynthetic labeling,” J Biomol NMR, vol. 48, no. 2, pp. 113–121, Oct. 2010, doi: 10.1007/s10858-010-9443-7.

We enclosed the file of manuscript with the marked changes from the reviewed version.

Reviewer 2 Report

S100A9 has been shown to co-aggregate with a-synnuclein and is involved in the amyloid-neuroinflammatory cascade in Parkinson’s disease. The work by Toleikis et al., follows on from earlier work that further characterizes the interaction between S100A9 and a-synnuclein using NMR spectroscopy and molecular dynamics simulations. However, what remains unclear is the functional importance of this interaction.

Other major concerns

1.    Previous studies have shown that S100A9 interacts with the α-syn C-terminus. Here the authors mentioned that the interaction is mediated by α-syn N-terminus. The authors tried to briefly explain the reason for the discrepancies in the discussion section. The authors should check the binding between the two partner α-syn and S100A9. They should check the interaction of S100A9 with N-terminal or C-terminal deleted α-syn to get clear idea of the interacting motif.

2.    S100A9 belongs to S100 protein family. Other members of that family specially S100A8 can form higher order oligomer as well as heterodimer. Is there any evidence about the interaction of other members of these family with α-syn?

3.    Can the authors comment on the morphology and the length of the filaments in presence or absence of S100A9?

4.    The authors should mention the percentage of sequence Identity and similarity between calmodulin and S100A9

5.    The result of ThT assay is not conclusive. It is not clear how S100A9 influence the kinetics of fibril formation.

6.    Liquid–liquid phase separation of α-Syn precedes its aggregation (Nature Chemistry volume 12pages705–716 (2020)). The factors that promote α-Syn aggregation, also promote its liquid–liquid phase separation and its subsequent maturation. Does S100A9 promote LLPS of α-Syn?

7.    Stating Kd in the “μM–mM range” (line 101) does not give any useful information as it is a board range. Either the authors can measure the Kd or omit the sentence

Minor concern

1.    Since the authors have performed all the studies with purified proteins, they should include the purification profiles (gel filtration and SDS PAGE) at least in Appendix A

2.    The authors should check the manuscript carefully, there are several grammatical mistakes throughout the manuscript.

Author Response

Please find our response shown in blue.

S100A9 has been shown to co-aggregate with a-synuclein and is involved in the amyloid-neuroinflammatory cascade in Parkinson's disease. The work by Toleikis et al. follows from earlier work that further characterizes the interaction between S100A9 and a-synuclein using NMR spectroscopy and molecular dynamics simulations. However, what remains unclear is the functional importance of this interaction.

The function of this complex is not known, but it might be that S100A9 can reduce the affinity of α-syn to interact with other proteins or lipid surfaces like vesicles or plasma membranes. Our data show that fibrils obtained by aggregating α-syn in the presence of S100A9 are less toxic to cell cultures. Therefore, we speculate that S100A9 interaction with α-syn has a protective function.

Other major concerns

1.    Previous studies have shown that S100A9 interacts with the α-syn C-terminus. Here the authors mentioned that the interaction is mediated by α-syn N-terminus. The authors tried to briefly explain the reason for the discrepancies in the discussion section. The authors should check the binding between the two partner α-syn and S100A9. They should check the interaction of S100A9 with N-terminal or C-terminal deleted α-syn to get clear idea of the interacting motif.

The previous study (Horvath et al., 2018) emphasized, that S100A9 interacts with the C-terminus of α-syn based on NMR data. However, if we analyze the figure describing the peak intensity change of each α-syn amino acid residue, we could see that the binding of S100A9 reduced the intensity of peaks also at the N-terminal part. Even looking at the HSQC spectrum in the mentioned study, it is clear that the S9 peak of α-syn disappears when S100A9 is added. This indicates that S100A9 binds to N and C-terminal parts when no calcium ions are present. It is known that calcium ions interact with the C-terminal part of α-syn, which might reduce the binding of S100A9. We observe a slight binding of S100A9 at the C-terminal part of α-syn from the HSQC spectrum, but this interaction is higher at the N-terminal part, as shown by our 19F and 15N-1H HSQC NMR data. Our sample contained calcium, which reduced the interaction at the C-terminal part. However, the NMR results allowed identification of each amino acid residue, which interacts from the side of both α-syn and S100A9 proteins. The deletions of C or N-terminal parts could confirm the findings of interacting regions. However, this experiment would take several months to test and is beyond the scope of our study.

I. Horvath et al., “Co-aggregation of pro-inflammatory S100A9 with α-synuclein in Parkinson’s disease: ex vivo and in vitro studies,” J Neuroinflammation, vol. 15, 2018, doi: 10.1186/s12974-018-1210-9.

2.    S100A9 belongs to S100 protein family. Other members of that family specially S100A8 can form higher order oligomer as well as heterodimer. Is there any evidence about the interaction of other members of these family with α-syn?

As far as we know, there are no publications about α-syn interaction with S100A8 or S100A8/S100A9 heterodimer. We agree that it would be a fascinating topic to study this interaction in the future.

3.    Can the authors comment on the morphology and the length of the filaments in presence or absence of S100A9?

Regarding your comment, we made the AFM analysis of fibril samples and added these results in the corrected manuscript version.

4.    The authors should mention the percentage of sequence Identity and similarity between calmodulin and S100A9

The sequence identity and similarity was 11% and 32%, respectively. Both proteins also have two EF-hand motifs and the 3D structures are similar. We added this to the manuscript.

5.    The result of ThT assay is not conclusive. It is not clear how S100A9 influence the kinetics of fibril formation.

The kinetics are too stochastic to find apparent differences in aggregation kinetics of our samples. There was an observation that the stochastic process was reduced in the presence of S100A9.

6.    Liquid–liquid phase separation of α-Syn precedes its aggregation (Nature Chemistry volume 12, pages705–716 (2020)). The factors that promote α-Syn aggregation, also promote its liquid–liquid phase separation and its subsequent maturation. Does S100A9 promote LLPS of α-Syn?

It could be that S100A9 would promote LLPS of α-Syn, but our aggregation conditions include 250-300 RPM shaking in the presence of a 3 mm glass bead, which most likely disturbs the LLPS.

7.    Stating Kd in the “μM–mM range” (line 101) does not give any useful information as it is a board range. Either the authors can measure the Kd or omit the sentence

We agree that this is a broad range of dissociation constant, but we thought this information could be helpful. We deleted this sentence, as we can not determine Kd precisely.

Minor concern

1.    Since the authors have performed all the studies with purified proteins, they should include the purification profiles (gel filtration and SDS PAGE) at least in Appendix A

We have added the gel filtration chromatogram of S100A9 and the SDS page gel image of purified S100A9 and α-syn. We did not use gel filtration for α-syn purification, but we added the ion-exchange chromatogram.

2.    The authors should check the manuscript carefully, there are several grammatical mistakes throughout the manuscript.

We have checked and corrected the mistakes.

The manuscript file enclosed is with the marks what was changed.

Round 2

Reviewer 2 Report

The authors have performed AFM experiments to calculate the length of the fibrils. However, there are few issues in the manuscript, that needs to be corrected. I recommend minor revision

1. The authors should use the greek letters uniformly throughout the manuscript (in case of alpha, beta)

2. It seems from figureA6C that S100A9 is not pure. Is it the protein obtained after gel filtration?

3. They can include the elution positions of the standards (clibration of the column) in the gel filtration profile figureA6A. Otherwise, it is difficult to get any idea about the size of the native protein.

Author Response

The answers are in blue

1. The authors should use the greek letters uniformly throughout the manuscript (in case of alpha, beta)

Thank you for this comment. We made the Greek letters more similar in the text and in the figures.

2. It seems from figureA6C that S100A9 is not pure. Is it the protein obtained after gel filtration?

We agree, that the protein is not 100 % pure, but the purity is approximately 90 %. We used ion exchange and gel filtration, the standard purification protocol for proteins without 6xHis-tag or other fused domains.

3. They can include the elution positions of the standards (clibration of the column) in the gel filtration profile figureA6A. Otherwise, it is difficult to get any idea about the size of the native protein.

Unfortunately we did not test the column with the standard proteins to determine the size of eluted S100A9. However, the protein after gel filtration should be monomer and some fraction could be a dimer also. The NMR spectra of S100A9 showed that the protein is not higher order than a dimer (more likely monomer) as the higher size of complex should not be visible in the spectrum.